# Versatile Use of Chitosan and Hyaluronan in Medicine [note 1]

**DOI:** 10.3390/molecules26041195

**Published:** 2021-02-23

**Authors:** Katarína Valachová, Ladislav Šoltés

**Affiliations:** Centre of Experimental Medicine, Slovak Academy of Sciences, Dúbravská Cesta 9, 84104 Bratislava, Slovakia; ladislav.soltes@savba.sk

**Keywords:** chitin, hyaluronic acid, polysaccharides

## Abstract

Chitosan is industrially acquired by the alkaline *N*-deacetylation of chitin. Chitin belongs to the β-*N*-acetyl-glucosamine polymers, providing structure, contrary to α-polymers, which provide food and energy. Another β-polymer providing structure is hyaluronan. A lot of studies have been performed on chitosan to explore its industrial use. Since chitosan is biodegradable, non-toxic, bacteriostatic, and fungistatic, it has numerous applications in medicine. Hyaluronan, one of the major structural components of the extracellular matrix in vertebrate tissues, is broadly exploited in medicine as well. This review summarizes the main areas where these two biopolymers have an impact. The reviewed areas mostly cover most medical applications, along with non-medical applications, such as cosmetics.

## 1. Introduction

Chitin, the second most abundant naturally occurring polysaccharide after cellulose, is the structural element in the exoskeleton of many animals, especially in crustaceans (snails, crabs, shrimps, etc.), mollusks, arthropods, the cuticles of insects, the scales of fish, and in the internal structure of invertebrates. Chitin can also be found in fungi, where it is the principal fibril polymer in their cell wall [1,2,3]. The negligible solubility of chitin is a limiting factor for its applications when the nonsolid form of this polymer should be a principal prerequisite [4]. Soft chitin has been widely used to immobilize enzymes, as desired in the food industry, such as in the clarification of fruit juices and processing of milk, when invertase or α and β-amylases are grafted on chitin. Chitin films and fibers have been used in medicine and pharmacy as storage materials for, e.g., wound dressings, and as drug releasing substances. Chitin has been used to prepare packings for affinity chromatography columns. Chitin-based materials have been applied in the adsorptive removal of heavy metals for water treatment [3].

The most known chitin derivative is chitosan, which rarely occurs in the environment; however, this polymer is smartly fabricated via the alkaline *N*-deacetylation of chitin. The product, chitosan, is classifiable as a linear polysaccharide composed of randomly distributed β-(1→4)-linked d-glucosamine (deacetylated unit) and *N*-acetyl-d-glucosamine (acetylated unit). Chitosan is soluble in acidic solutions, wherein it becomes protonized [1,5]. Chitosan bears several advantageous properties, such as biocompatibility, progressive degradability, absence of toxicity, physiological inertness, remarkable affinity to proteins, hemostaticity, lack of allergenicity, and antibacterial activity. In contrast to most other biopolymers, chitosan in acidic aqueous solutions has a positive electrical charge due to its protonized glucosamine groups. This charge resulted in chitosan’s electrostatic attachment to most body tissues which bear negatively charged surface matrices [5,6]. Chitosan and several of its derivatives have been widely used in cosmetics and in the field of medicine, including biomaterials for tissue-engineered scaffold and tissue repair, wound dressings, and biochemical separation systems [1,7,8,9,10,11,12,13,14,15]. Moreover, chitosan serves as a food preservative. Both chitin and chitosan are of high importance in wastewater treatment. They can be used as coagulating and flocculating agents for polluted wastewaters, in heavy metal or metalloid adsorption, e.g., Cu(II), Cd(II), Pb(II), Fe(III), Zn(II), and Cr(III), for the removal of dyes from industrial wastewater, e.g., from textile wastewaters, as well as for the removal of other organic pollutants such as organochloride pesticides, organic oxidized compounds, or fatty and oil impurities [13]. Chitosan can also participate in ruminants’ fermentation process [13,16]. Chitosan, due to its positive charge, efficiently binds negatively charged molecules, including fatty acids, lipids, and bile acids, properly excreting these molecules from the body. Thus, as claimed, chitosan considerably reduces the level of cholesterol in the blood [4,17,18]. Chitosan oligosaccharides, which usually have a degree of deacetylation less than approx. 50% and an average molar mass less than 10,000 Da, are very water soluble, resulting in solutions of low viscosity; due to these characteristics, chitosan oligosaccharides are favorably applied in biomedicine, contrary to high-molar-mass chitosan [10,16,19]. Chitosan and chitosan oligosaccharides can be used to attenuate metabolic syndrome by altering the microbiota in the gut, decreasing the risks of cardiovascular disease, and reducing inflammation in the intestinal mucosa [20].

Hyaluronan (HA) can be grouped among the glycosaminoglycans, whose linear chain comprises alternating β-1,4-glucuronic acid and *N*-acetylglucosamine [21,22]. HA is a major component of the extracellular matrix, where it promotes cell proliferation and migration of fibroblasts and keratinocytes [23]. The average 70 kg adult human body contains approximately 15 g of HA, of which one third is turned over every day [24]. This biopolymer is really outstanding among the natural hygroscopic (macro)molecules; during the process of water absorbance, high-molar-mass HA molecules swell in volume up to 1000 times in the form of polymer powder [24,25,26,27]. The biological effects of HA depend on their molar masses. HA of molar mass 0.4 to 4.0 kilodalton (kDa) functions as an inducer of heat shock proteins, and such lower sized molecules has a non-apoptotic property. HA of molar mass equaling to 6–20 kDa possesses immunostimulatory and angiogenic properties. When the molar mass of HA reaches 20–200 kDa, the macromolecules take part in biological processes such as embryonic development, wound healing, and ovulation. By contrast, high-molar-mass HA (>500 kDa) has anti-angiogenic activity, and can function as a space filler and a natural immunologic depressant [28]. The HA biopolymer provides a wide range of pharmacological activities, including anti-inflammatory, wound healing and tissue regenerating, immunomodulatory, anti-cancer, anti-proliferative, anti-diabetic, anti-aging, skin repairing, and cosmetic properties [27,29].

### 1.1. Viscosupplementation

Natural biopolymers such as HA and chitosan are among the most investigated biomaterials for cartilage regeneration. One promising approach to repair cartilage is the transplantation of autologous chondrocytes or stem cells into cartilage defects using a natural or synthetic scaffold. To perform the cartilage repair function, the scaffold must be biocompatible with the cartilaginous tissue, biodegradable, nontoxic, and non-immunogenic, which are characteristics attributed to both HA and chitosan biopolymers [11,30].

Thus, while intra-articular chitosan appears to prevent cartilage destruction and may induce a reparative process due to cell attachment, it has also been associated with an inflammatory process, and exposure to chitosan has been linked to autologous blood coagulation. In addition, chitosan has been shown to improve joint lubrication when injected intraarticularly in humans. When used together with a chemical holmium, the latter element is advantageously used for diagnostics purposes [6]. Chitosan-alginate beads could be used as a carrier for cell transplantation, particularly to repair cartilage defect. These beads produced less inflammatory and catabolic mediators and maintained the synthesis of cartilage specific matrix components [11].

HA, within synovial fluid, plays an important role in the protection of articular cartilage and the transport of nutrients to cartilage. Degradation of HA by either hyaluronidases or reactive oxygen species leads to a lower size of native high-molar-mass HA, and thereby the viscosity of synovial fluid decreases and cartilage can degenerate [29,31]. Hyaluronan plays a role in the hydration and lubrication of joints due to its space filling capacity and by means of its mechanical shock-absorbing properties [26,32,33]. HA has either a visco- or an elastic behavior as a function of the frequency [29]. Some further functions of HA are summarized in Table 1.

Moreover, since HA can also be used as a scaffold in bone regeneration, it is a proper material for osteochondral tissue engineering. Hyaluronan is most generally used in the cross-linked, modified form or as a composite with other materials such as platelet-rich plasma, fibrin, chitosan, collagen, or alginate due to these forms having better mechanical properties than the linear HA [35]. Unterman et al. [30] examined hyaluronic acid-based scaffold to repair articular cartilage.

With cartilage tissue engineering, chondrocytes can be encapsulated into hydrogel networks in order to treat the damaged cartilage tissue. One such hydrogel is called Gel-One, which is composed of Gel-200 product—a cross-linked hyaluronate hydrogel. Another new product, Hyajoint Plus, was shown to produce a longer lasting and stronger effect on pain than compared to Synvisc-one, a viscoelastic solution which has been used by many physicians in intraarticular injections [36]. One currently used HA commercial product is Ostenil^®^, which must be injected once a week for three to five weeks to get an effect lasting around six months [37]. To minimize the number of intraarticular injections, a smart approach has been patented [38]; a mixture of two self-associating HA derivatives is applied into the joint where a viscoelastic gel is formed [39,40].

Bovine chondrocytes were encapsulated with the cross-linked hydrogel made of *N*-succinyl-chitosan and hyaluronic acid, which supported cell survival, and the cells retained chondrocytic morphology [19]. Chemically modified hyaluronic acid–chitin and hyaluronic acid–chitosan materials were reported to be osteoinductive and exhibited rapid degradation and neovascularization in vivo [18]. Wang et al. [41] showed that HA/chitosan nanoparticles for the intra-articular delivery of curcuminoid can decrease chondrocyte apoptosis in rats with knee osteoarthritis through repression of the NF-κB pathway.

### 1.2. Viscoprotection

Chitosan has excellent ocular tolerance. Ophthalmic formulations based on chitosan are easy to manufacture and exhibited an excellent tolerance when topically administered onto the cornea, since chitosan solutions have shown pseudoplastic and viscoelastic properties [16,42]. Most of the reports on the topical ocular administration of chitosan solutions refer to concentrations in the range of 0.5–5% and a molar mass higher than 70–100 kDa. An alternative way to modulate the viscosity and viscoelastic behavior of chitosan solutions could be through the incorporation of other hydrophilic polymers that are known to interact with chitosan, e.g., hyaluronic acid [1]. Ocular drug delivery of nanoparticulate prulifloxacin in situ gel based on thiolated chitosan with mean particle size 16 nm and efficiency of 80% drug entrapment showed gelation at pH 7.2 ± 0.2. This viscoprotective material had excellent mucoadhesive and non-irritant properties, and exhibited sustained drug release over the cornea [43].

Acharya et al. [44] prepared gellan gum and chitosan-based in situ gel of timolol maleate, which was shown to be a good alternative for eye drops, as this gel sustains the drug release for a prolonged period of time, thus reducing the number of applications of the eye drops. Barwal et al. [45] showed that using ultra-small chitosan particles loaded with brimonidine was efficient in treating patients with glaucoma. DeCampos et al. [46] showed that chitosan nanoparticles are able to interact and remain associated to the ocular mucosa for extended periods of time, thus being promising carriers for enhancing and controlling the release of drugs to the ocular surface. Chitosan can interact with the negative charges of the ocular mucus, thereby enhancing bioadhesion [47]. Various drugs have been loaded on chitosan-based thermosensitive hydrogels for the treatment of ocular diseases. For example, latanoprost was loaded on chitosan- and gelatin-based thermosensitive hydrogel for controlling ocular hypertension [48].

Overall, chitosan-based formulations appear to be less viscous than those based on traditional viscolizers such as hyaluronan. HA is widely used in ophthalmology due to its viscosupplementing properties and its natural presence in the cornea, the sclera, and the vitreous body. Hyaluronan-derived products have been used to maintain the structure of the eye’s capsule, mainly in the phacoemulsification (removal of the eye’s vitreous humor) step of cataract surgery, but also in other surgical procedures such as corneal transplantation, intraocular lens implantation, and vitreoretinal, anterior segment, and glaucoma surgery. Some biocompatible materials were introduced in the ophthalmic practice for the vitreous humor replacement after vitrectomy using a new crosslinked HA through adipic acid dihydrazide bridges [25]. It has been demonstrated that treatment with eye drops containing HA reduces oxidative stress in the conjunctiva of patients with dry eye disease [49,50]. Silvani et al. [51] investigated new artificial eye drops based on arabinogalactan and hyaluronic acid; the authors showed that this combination involved the reduction in uric acid and reactive oxygen species by 38% and 62%, respectively, in vitro; therefore, the use of this preparation may help to treat dry eye syndrome. The first ophthalmic viscosurgical device containing HA was approved by the FDA in 1980 and is still marketed under the trademark Healon^®^. Moreover, HA is an active ingredient of many eye drops, such as DropStar^®^ by Bracco and Lubristil^®^ by Eyelab, which hydrate the ocular surface and improve the quality of vision, and therefore are useful in the treatment of diseases such as dry eye syndrome and in increasing the comfortability of contact lenses use [52]. Griffith et al. [53] showed that in male New Zealand rabbits with corneal chemical burn, a crosslinked thiolated HA film significantly decreased the areas of corneal opacity within 14 days postoperative treatment. Durrie et al. [54] accelerated the reepithelization of corneal defects in 39 patients suffering from photorefractive keratectomy using thiolated carboxymethyl hyaluronic acid gel. Stodolak-Zych et al. [55] used a membrane consisting of chitosan, collagen, and hyaluronic acid in the presence of polyethylene oxide for the treatment of corneal ulcers.

### 1.3. Drug Delivery and Gene Therapy

The drug delivery systems offer many advantages in therapy, which include: (a) reduced drug toxicity, (b) increased therapeutic index of the drug, and (c) prevention of frequent, expensive, and unpleasant dosing. Chitosan-based porous scaffolds and its composite nanoparticles have emerged as a 3D matrix in a prolonged topical delivery of several drugs [56]. Chitosan and its derivatives may be used as solutions, gels, tablets, capsules, fibers, films, and sponges [18,56,57]. Consequently, such drug formulations may be administered orally, ocularly, nasally, vaginally, buccally, parenterally, intravesically, and transdermally. Moreover, they may serve as implants for drug delivery in both implantable and injectable forms. The advantageous properties of chitosan resulted in the development of vaccine delivery [4,18].

A novel chitosan/acellular dermal matrix stem cell delivery system was developed and demonstrated its great potential in stem cell therapies in wound healing [58]. The application of nanosized materials is a promising strategy for topical drug delivery due to their enhancing effect on drug percutaneous transport across the *stratum corneum* barrier. The authors prepared polymeric micelles made from hydrophobized hyaluronic acid and examined these micelles for skin delivery. These micelles were found to be stable in cream formulations and thus they have great potential for topical applications in the cosmetic and pharmaceutical industry [59]. All-trans retinoic acid (ATRA) was grafted to HA via esterification. The chemical structure of HA-ATRA and its degradation products was elucidated using NMR spectroscopy, SEC-MALLS, and gas chromatography-mass spectrometry. ATRA did not lose its biological activity after conjugation, as demonstrated by gene expression: the derivative was able to penetrate across the *stratum corneum*. Besides this phenomenon, HA-ATRA downregulated the expression of anti-inflammatory IL 6 and IL 8. HA-ATRA would be used for transdermal drug delivery or cosmetics [60]. The use of chitosan as a potential excipient in pharmacy for drug delivery produced via direct tablet compression was examined. Chitosan, *N*-cinnamyl substituted *O*-amine functionalized chitosan, and microcrystalline cellulose (MCC) were formulated, alongside with acetaminophen as the active pharmaceutical ingredient, and magnesium stearate (Mg-St) as a lubricant in formulated blends. A control blend of MCC, acetaminophen (20 wt.%), and Mg-St (0.5 wt.%) was studied alongside two chitosan-bearing blends, containing 20 wt.% chitosan and 20 wt.% cinnamyl-chitosan separately. The particle size, shape, and morphology of the raw powders were studied, along with flowability of both raw powders and formulated powder blends. Blends containing chitosan and cinnamyl-chitosan possessed good compaction properties with high elasticity due to their large particle sizes, and showed excellent dissolution properties, releasing >80% acetaminophen within 30 min. With good mechanical strength and superior drug delivery performance, in addition to its enhanced antibacterial and antioxidative effect gained through chemical modification, cinnamyl-chitosan exhibited a potential to be used as a new cost-effective pharmaceutical excipient in direct compression tableting [61].

HA-chitosan nanoparticles have the potential to serve as a reliable drug delivery system to topically treat ocular surface disorders [62]. Biocompatible and biodegradable HA-coated chitosan nanoparticles were developed to encapsulate a chemotherapeutic drug (5-fluorouracil) to enhance drug accumulation in tumor cells and to improve the agent’s antitumor efficiency by offering targeted drug delivery via CD44 [63]. A drug containing poly(ethylene glycol)/chitosan microspheres with glutaraldehyde as the cross-linking agent was also reported [14]. Chitosan nanoparticles as potential drug delivery carriers have been widely studied in cancer treatment, as well as in the early diagnosis and detection of tumor biomarkers in vivo and in tumor-targeted therapy [64]. Gene therapy is a promising strategy for treating rare and fatal diseases, where a critical step is the successful delivery of genes to a site of damage. However, concerns about immunogenicity and toxicity are the critical obstacles against the world-wide use of effective viral systems. Therefore, nonviral vectors are considered as appropriate alternatives to viral vectors. One option is to use chitosan to create nonviral gene delivery vectors [18,65] as promising delivery biomaterials in gene therapy. The complexes of chitosan–DNA are easy to prepare and are more effective compared to the commonly used delivery systems. These complexes are reported to be transfected into various cell types, such as human embryonic kidney cells, cervical cancer cells, primary chondrocytes, and fibroblast cells [18]. Zhou et al. [66] demonstrated the feasibility of using a non-viral gene delivery system composed of chitosan and HA for the expression of CrmA gene in a rat model of osteoarthritis with attenuated cartilage damage and synovial inflammation. Chitosan in the form of nano-particles and resorbable films can be utilized to deliver drugs (such as metronidazole, chlorhexidine, or nystatin) to periodontal tissues in situ, against fungal infections and oral mucositis [67]. Moreover, chitosan can form complexes with negatively charged genes easily due to its abundant amine groups. However, clinical translation of chitosan-based gene delivery carriers is still unsatisfactory due to several challenges, such as poor water solubility, charge deduction at physiological pH, and poor targeting capability [68].

Chitosan in combination with disodium α-d-glucose-l-phosphate has been used for ocular drug delivery system. A novel copolymer, poly(*N*-isopropylacrylamide)-chitosan, was investigated for its thermosensitive in situ gel-forming properties and potential utilization for ocular drug delivery. Another novel thermosensitive hydrogel was made by using chitosan and glycidyltrimethylammonium chloride and named as *N*-[(2-hydroxy-3-trimethylammonium) propyl] chloride chitosan [69]. Mucoadhesive chitosan-coated cationic microemulsion has been used as a carrier of dexamethasone for ocular delivery to treat ocular diseases [48]. Selvaraj and Nirajmathi [70] showed that acyclovir loaded chitosan nanoparticle suspension appears to be promising for effective management of ocular viral infections. The efficacy of ocular formulations is limited by poor corneal retention and permeation, resulting in low ocular bioavailability. Taghe and Mirzaeei [71] showed that mucoadhesive chitosan/sodium tripolyphosphate and chitosan/sodiumtripolyphosphate-alginate nanoparticles demonstrate prolonged topical ophthalmic delivery of ofloxacin. To date, the rapid clearance from the ocular surface has been a huge obstacle for using eye drops to treat glaucoma, since it has led to a short preocular residence time and low bioavailability. The novel nanoparticles were designed for a topical ophthalmic controlled drug delivery system through intercalating the betaxolol hydrochloride into the interlayer gallery of Na-montmorillonite and then further enchasing chitosan nanoparticles [72]. In addition, chitosan–miRNA complexes were investigated to target cystic fibrosis cells. Supramolecular chitosan-based nanostructures for gene therapy can be prepared by simple complexation, ionic gelation using crosslinkers and adsorption of DNA/siRNA onto the surface of preformed chitosan nanoparticles [68,73].

Chitosan-*N*-acetylcysteine has been approved on the market as eye drops under the name Lacrimera, with increased mucoadhesive properties. Chitosan-coated liposomes, called chitosomes, increased ocular retention with decreased metabolism of drug substances. Coating liposomes with quaternary ammonium chitosan derivatives such as *N*-trimethylchitosan reduce particle aggregation due to steric stability and increases mucoadhesiveness. The association of chitosan and gelatin has been beneficial in the preparation of contact lenses [16,56]. Hu et al. [74] examined a chitosan/hyaluronic acid multilayer of contact lenses, which were loaded with norfloxacin and timolol. Such a coating resulted in increasing the hydrophilic character of lenses and the water retention and a reduction in the deposition of proteins. Table 2 summarizes various chitosan-based drug delivery systems.

### 1.4. Dentistry

The use of chitosan has been proposed in all fields of dentistry, including preventive and/or conservative dentistry, endodontics, surgery, periodontology, prosthodontics, and orthodontics. Daily use of chitosan to rinse the oral cavity (polymer size related to 6 kDa mean molar mass, deacetylation degree approx. 40%, polymer concentration in the rinsing solution 0.5% *w*/*v*) was effective in reducing dental plaque formation and the count of salivary bacteria [5,19].

Chitosan can be used to repair tooth enamel, for coating dental implants, and in stem-based regenerative therapeutics [67]. Another use of chitosan in dentistry is related to avoiding dental abrasion: a toothpaste containing chitosan (Chitodent^®^) was compared with a propolis-containing toothpaste (Aagaard^®^), a fluoridated (500 ppm) toothpaste (Elmex^®^), and a control group without treatment. Both propolis and chitosan toothpaste showed a lower average brushing abrasion value on healthy tooth surfaces than Elmex^®^ toothpaste. Moreover, chitosan gels containing herbal extracts showed a dental plaque reduction of 70%, and up to a 85% reduction in bacterial count. Chitosan microparticles containing NaF produced by spray drying exhibited bioadhesive properties to teeth, thus acting as a proper fluoride reservoir [19]. Salts of chitosan added to toothpaste disguise the unpleasant taste of silicon oxide. Complexes of chitosan and fluoride microparticles increase fluoride absorption and the protection of cavities. Endodontic cements based on chitosan reduce inflammation and support bone regeneration [18]. Diolosa et al. [75] prepared a modified chitosan with methacrylic acid (Chit-MA70) on 16% of the amino groups and investigated the effect of Chit-MA70 on the durability of adhesive interfaces to improve the clinical performance of dental restorations. The results show that the presence of methacrylate moieties and residual positive charges on the polysaccharide chain allowed Chit-MA70 to covalently bind to the restorative material and electrostatically interact with demineralized dentin.

Hyaluronan is used by dentists to serve as an adjuvant in tissue reparation and traumatic processes. HAs are generally used as antiseptics, which is beneficial to decrease bleeding. The biopolymer HA is used in conditions of traumatic, degenerative, or inflammatory temporomandibular joints, since HA decreases pain (favoring nutrition to avascular areas of the condylar cartilage and disc). Other reported uses of HA are in maxillofacial surgery, orthopedics, and orthognathic surgery. In periodontal therapy, HA has been employed in gingivitis, recessions, periodontal pockets, grafts, and implants [76]. Coimbra et al. [77] investigated the usage of HA/chitosan scaffolds for dental pulp regeneration. Palma et al. [78] investigated the newly formed tissues after regenerative endodontic procedures in dogs using either a blood clot, hyaluronan:chitosan scaffolds, or pectin:chitosan scaffolds. The greatest amount of mineralized tissue inside the canal was observed in the group using the blood clot. In contrast, the addition of hyaluronan:chitosan scaffolds, or pectin:chitosan scaffolds in dogs did not support the formation of new mineralized tissues along the root canal walls or the histological evidence of the regeneration of a pulp-dentin complex.

### 1.5. Cosmetics

Chitosan and its derivatives have been included in a wide variety of hair products, such as shampoos, rinses, permanent wave agents, hair colorants, styling lotions, hair sprays, and hair tonics [10,19,79]. Chitosan blended with HA and collagen produced films, which, as it was shown, increased hair thickness and improved the mechanical properties of hair [80]. Chitosan, microcrystalline chitosan, and quaternized chitosan were added to shampoos and hair sprays due to their film forming activity and moisturizing effect. Chitosan with glycerol was included as a component in liquid hair strengtheners and hair sprays due to its improved solubility and film forming capacity. Chitosan argininamide is so far the only chitosan derivative identified as a skin cleaner. Chitosan derivatives containing fatty acids such as lauramide, succinamide, lauroyl glycinate, and hydrolyzed ferulyl linoleate have been described as stabilizers of cosmetic products [19]. HA-based cosmetics such as Fillerina^®^ (Labo Cosprophar Suisse) claims to restore skin hydration and elasticity [52].

High-molar-mass chitosan decreases the loss of trans-epidermic water and increases the humidity of skin, resulting in the preservation of the skin softness and flexibility. Moreover, chitosan shows additional advantages in protection against sun light. There are marketed creams enriched with vitamin E containing water-soluble chitosan of different mean molar masses and degree of deacetylation. Moreover, high-molar-mass chitosan can help to reduce the transpiration of gases/vapors through the skin [10].

Numerous chitosan-based products for cosmetic use are commercially available, such as Curasan™, Hydamer™, Zenvivo™, Ritachitosan^®^, and Chitosan MM222. As indicated, cosmeceuticals (e.g., Chitoseen™-K) are cosmetics with numerous pharmaceutical benefits. They can be applied as creams, lotions, and ointments [18].

### 1.6. Tissue Engineering and Wound Healing

Tissue engineering includes the development and manipulation of laboratory-grown cells, tissues or organs that could replace or support the function of defective or injured parts of the body [18,79]. This technique commonly needs to employ three dimensional supports for initial cell attachment and subsequent tissue formation [56]. Special attention on chitosan has been paid for the repair of articular cartilage [17,56,79]. The polycationic properties of chitosan are being developed for use in biosensors by immobilizing enzymes, in wound dressings to induce cell migration and proliferation at the wound site, and in tissue engineering as a scaffold [2]. Microporous chitosan/calcium phosphate composite 3D scaffolds were synthesized and characterized for tissue engineering, whereas chitosan provides a scaffold form and calcium phosphates encourages osteoblast attachment and strengthens the scaffold [79]. Moreover, chitosan-chondroitin sulphate sponges for bone regeneration were formed. Chitosan-calcium alginate capsules were made to develop an artificial pancreas for the treatment of diabetes mellitus [17].

Chitosan has similar structural characteristics to glycosaminoglycans found in the extracellular matrix of several human tissues [56], and is a prominent dressing in wound healing owing to the fact that it not only prevents bacterial infection in wounds, but also enhances healing and leads to less scarring. Furthermore, chitosan can also deliver a therapeutic dose to a local wound, e.g., fibroblast growth factor 2, which stimulates angiogenesis by activating capillary endothelial cells and fibroblasts. To overcome the poor solubility of chitosan in water, its more rapid in vivo depolymerization and degradation, its hemo-incompatibility, and also its insufficient antimicrobial properties, chitosan derivatives have been synthesized as novel scaffold materials for tissue engineering. The major commercial applications of chitosan are in wound healing, which fulfills the characteristics as follows: no toxicity and no irritability, sufficient mechanical strength to prevent forming wrinkles, good moisture and air/water vapor permeability, biodegradability, and antibacterial properties against wound infection [18,81].

On the market, there are commercially available various forms of chitosan-based wound dressing materials, such as HemCon^®^ Bandage, ChitoGauze^®^ PRO, ChitoFlex^®^PRO, ChitoSam™, Syvek-Patch^®^, Chitopack C^®^, ChitopackS^®^, Chitodine^®^, ChitosanSkin^®^, TraumaStat^®^, TraumaDEX^®^, and Celox™ [18]. Chitosan tends to support tissue healing by encouraging blood coagulation. The improved healing might also be related to the increased permeability of cell membranes in contact with chitosan, where the extent of contact is dependent primarily on the presence of particles of a proper size [6]. Oh et al. [82] reported the preparation of chitosan-based film containing a nosteroidal antiinflammatory drug with analgetic and antipyretic properties—ketoprofen. As claimed, this composite film bears excellent mechanical properties. Ishihara et al. [83] prepared a photocrosslinkable chitosan derivative with beneficial effects on skin wound healing in mice. Applications of nano-colloidal silver in conjunction with chitosan as primary bioactive dressings in managing diabetic foot ulcers patients was safe and helped to enhance wound healing [84]. Yang et al. [85] suggested a new strategy in which injectable thermosensitive chitosan/collagen/β-glycerophosphate hydrogels were combined with three-dimensional mesenchymal stem cell spheroids to facilitate chronic wound healing through enhanced vascularization and paracrine effects.

Chitosan’s cationic charge property facilitates its binding with red blood cells: it allows rapid clotting of the blood, and this biopolymer has gained regulatory approval in the USA for use in bandages and hemostatic agents. In addition, chitosan modulates the functions of inflammatory cells and subsequently promotes granulation and tissue organization. As a component of semipermeable biological dressings, chitosan maintains a sterile wound exudate beneath a dry scab, prevents dehydration and contamination of wounds, and optimizes conditions for healing [8,12]. Chitosan/nanoselenium biofilm resulted in a significant healing of infected skin wounds in rats [86]. Breder et al. [87] showed a beneficial effect of topical treatment with a chitosan-alginate membrane on acute skin wounds of hyperglycemic mice. The authors developed a stable chitosan hydrogel with incorporated flavonoid fraction, isolated from a *P. edulis Sims* leaf extract. The multicomponent hydrogel represents a semi-solid form, which resulted in forming an excellent film when applied to the skin of rats [88]. Ojeda-Martínez et al. [89] prepared and characterized chitosan thin films with silver nanospheres. Azad et al. [90] showed the beneficial effect of chitosan membranes at the skin-graft donor site in patients. The effects of chitosan membranes were successfully exploited on skin ulcers in patients [91]. Nordback et al. [92] proved the beneficial effects of chitosan membranes in cutaneous wounds in rats.

Some HA-based materials are commercially available for the treatment of deep dermal lesions such as non-healing ulcers, deep II and III degree burns (Hyalograft^TM^ 3D), for autologous skin replacement (Laserskin^®^), for autologous fibroblasts, and for keratinocyte replacement (Tissuetech^®^). Other commercially available HA-based products on the market are Hyalofill-f^®^, Hyalofill-R^®^, Connettivina^®^, Connettivina Plus^®^, Jaloskin^®^, and Hyalomatrix^®^ [25]. Hyiodine^®^ (Contipro) is a patented complex of 1.5% sodium hyaluronate, produced through a process of bacterial fermentation, which was developed for the treatment of skin wounds [93].

Membranes composed of chitosan and HA were examined and characterized by numerous investigators [94,95,96,97,98,99,100]. There are several papers which reported the use of membranes or scaffolds composed of chitosan and HA in the presence of another component, such as gelatin [101,102], alginate [103], collagen I [104], arginine derivatives [105], or polycaprolactone [106].

In our previous studies, we prepared chitosan/HA composite membranes loaded with various drugs, e.g., with edaravone, which have beneficial effects in treating wounded rats [107]. In another study, Tamer et al. [108] prepared and characterized membranes composed of chitosan, HA, and MitoQ: the drug MitoQ had a beneficial healing effect on both the structure of membranes and their application on skin wounds of rats and ears of ischemic rabbits. Similarly, tiopronin and captopril added to chitosan/HA membranes were potent to facilitate the healing of lacerations in ischemic ears of rabbits [109]. The chitosan/HA composite membranes loaded with glutathione (GSH) were shown to be more beneficial in the treatment of skin wounds in rats than in untreated rats and rats treated only with membranes without GSH [110]. Ergothioneine, histidine, and hercynine added individually to chitosan/HA membranes also contributed to a more rapid healing of skin lacerations in ischemic rabbits [111]. Hassan et al. [112] published a paper where the free-radical scavenging capacity of phosphatidylcholine dihydroquercetin was demonstrated in vitro and the results of the in vivo experiments in rats showed a beneficial effect of this substance when added to the chitosan/HA membranes. Soltes et al. [113] patented composite membranes containing a smart-released cytoprotectant targeting the inflamed tissue.

Table 3 and Table 4 describe chitosan modifications and the characteristics of commercial chitosan preparations.

## 2. Conclusions

Both chitosan and hyaluronan are polysaccharides of similar structure, which can be applied widely, individually or together, in medicine, especially in the treatment of osteoarthritis, in ophthalmology, skin wound healing, cosmetics, and dentistry. Moreover, chitin and especially chitosan have beneficial effects in other fields, such as the food industry, photography, wastewater treatment, and the chemical industry. In recent years, there has been a growing interest in investigating chitosan in gene therapy and tissue engineering. The most significant impact of chitosan in medicine is in the treatment of hard-to-heal wounds such as leg ulcers, burns, and decubites. Our research is focused especially on the preparation and characterization of membranes made of both polysaccharides. Chitosan is well known for its ability to form membranes. There are numerous papers reporting on the effects of chitosan in the form of films, scaffolds, fibers, and membranes. One possibility how to improve the chemical-physical properties of membranes is to introduce hyaluronan. Moreover, a third component, such as a drug/antioxidant with a sustained release to the skin wound, could have a great benefit in the application of chitosan/HA membranes for healing skin wounds.

## Figures and Tables

**Table 1 molecules-26-01195-t001:** Functions of HA in the synovium, the articular cartilage, and the extracellular matrix of the articular joint reviewed in reference [34].

Functions of HA in Articular Joints
Enhancing metabolism of chondrocytes
Prevention of the degradation of proteoglycans and collagen in the extracellular matrix
Inhibition of degeneration of chondrodrocytes
Protection of chondrocytes against apoptotic death

**Table 2 molecules-26-01195-t002:** Chitosan-based drug delivery systems prepared by different methods reviewed in reference [18].

Form	Drug	Method
Nanoparticles	Insulin, cyclosporin A	Emulsion droplet
		Ionic gelation
		Coalescence Coacervation/precipitation Reverse micellar method
Gels/hydrogels	Caffeine, lidocaine	Cross-linking reactions
	Insulin	Capsule shell
Beads	Bovine serum albumin, salbutamol	Coacervation/precipitation
	Trypsin, testosterone	Wet casting from salt solutions
Microspheres	Diclofenac, aspirin, 5-fluorouracil	Sieving method Water-in-oil emulsion
		Spray-drying Coacervation/precipitation Cross-linking
	Triamcinolone acetonide	Freeze drying
		Reactions in supercritical fluids
Tablets	Salicylic acid, diclofenac	Matrix coating

**Table 3 molecules-26-01195-t003:** Chitosan modifications for wound healing dressings reviewed in references [14,114].

Chitosan Derivatives	Characteristics
Carboxymethyl chitosan	The most widely explored derivative of chitosan with enhanced water solubility at pH > 7.
Carbohydrate branched chitosans	These derivatives are water soluble. Carbohydrates can be grafted onto the chitosan backbone at the C-2 position by reductive alkylation, which is essential since they are recognized by the corresponding specific lectins and thus could be applied for drug targeting.
Alkylated chitosan	Very essential as an amphiphilic polymer based on polysaccharides, enhances the stability of the interfacial film, cationic surfactant adsorbed on the alkyl chain grafted on chitosan, which promotes its solubilization.
Thiolated chitosan derivatives	Thiourea increases the antibacterial properties of chitosan derivative.
Trimethyl ammonium chitosan	Water soluble over all the pH range, is obtained by quaternization of chitosan, bearing good flocculating and antistatic properties.
*N*-Methylene phosphonic chitosans	Have good complexing efficiency for cations such as Ca(II), and transition metals (Cu, Zn, etc.). The complexation provides corrosion protection for metal surfaces. These derivatives were also modified and grafted with alkyl chains to obtain amphiphilic properties.
*N*-Succinyl chitosan	An amphiprotic derivative containing amine, hydroxyl, and carboxyl groups. It has excellent physical, chemical and biological properties required for biomedical applications.
Chitosan-grafted copolymers	One of the most explored derivatives is polyethylene glycol grafted chitosan, which has the advantage of being water soluble, depending on the degree of grafting.

**Table 4 molecules-26-01195-t004:** Commercial chitosan preparations and their characteristics reviewed in references [14,18,114].

Commercial Chitosan Preparations	Characteristics
Chitipack P^®^ Eisai Co	Swollen chitin dispersed in polyethylene terephthalate, favors early granulation tissue formation.
Chitipack S^®^ Eisai Co	Sponge-like chitin obtained from squid. Favors early granulation tissue formation, no retroactive scar formation. Suitable for traumatic wounds and surgical tissue defects.
Tegasorb^®^ 3M	Chitosan particles will swell while absorbing exudate, forming a soft gel. A layer of waterproof Tegaderm^®^ dressing covers a hydrocolloid. Suitable for leg ulcers, sacral wounds, chronic wounds.
Chitoflex^®^	HemCon chitosan-based is antibacterial and biocompatible. It combines strongly to tissue surfaces and forms a flexible barrier, which can seal and stabilize the wound. For stuffing into a wound track to control severe bleeding.
Chitoseal^®^ Abbott	Chitosan-based. Good biocompatibility and hemostatic function. For bleeding wounds
Chitopack C^®^ Eisai	Cotton-like chitosan. It repairs body tissue completely, rebuilds normal subcutaneous tissue and regenerates skin regularly.
Chitopoly^®^ Fuji	Chitosan and polynosic Junlon polyacrylate for preparing antimicrobial wears. For preventing dermatitis.
HemCon^®^ Bandage	Engineered chitosan acetate preparation designed as a high-performance hemostatic dressing.
Reaxon^®^ (Medovent, Germany)	Chitosan-based nerve conduit which is resistant to collapse and helps to avoid the undesired drawbacks of autografts. This hydrogel is bioactive (supports nerve regeneration equivalent to the autograft), biocompatible (prevents irritation and inflammation), antiadhesive (inhibits scar tissue and neuroma formation), and antibacterial (prevents infection).
ChitoSeat™	A family of chitosan-based hemostatic sealants that is suitable for surgical hemorrhage of hard and soft tissue.
Beschitin^®^, Unitika (Osaka, Japan)	A non-woven fabric manufactured from chitin filaments is also commercially available in Japan. It is recommended for the successful and fast healing of burns, skin abrasions, postoperative wounds, bed sores, ulcers and several other injuries.

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
