# Peer review of "Versatile Use of Chitosan and Hyaluronan in Medicine"

_molecules, 2021, doi:10.3390/molecules26041195_

Round 1

Reviewer 1 Report

Overview and general recommendation:

Chitosan and HA are molecules of high impact worldwide. They both have an impact in several industries and medicine. The present paper reviews both molecules' impact in ophthalmologic, drug delivery, dentistry, cosmetic, and wound-healing areas with commercial examples, making the manuscript of colossal importance for the community.  It is very professionally written, understandable and references are recent. However, some suggestions can be attended to improve the manuscript's quality. Some schemes could also be added to soften the reading and improve the understanding of the concepts and applications.

The manuscript's length is enough to make the reading pleasant. However, future trends or special remarks section could highlight the chitosan and HA applications' future. Conclusions must also be related to the paper content highlighting the most critical aspects reviewed.

Abstract: Line 13, please focus on the described areas in the paper. Not all the chitosan and HA application areas were described. For example, tissue engineering is not described inside the paper. Also, gene delivery is barely described. Both areas are also highly imprinted by chitosan. There are also papers describing chitosan treatment of the metabolic syndrome.

Line 42: Please provide examples of negatively charged surfaces.

Line 46: It is essential to explain better how chitosan application in fermentation and water treatment processes.

Lines 83-85 need a reference to better support the statement.

Table 1. It should be added a column with several references of studies demonstrating those HA functions and properties.

Line 218: What is CD44? Explain

Line 229: It is important to highlight what is the significant relevance of the CrmA gene.

Line 230: It should be nanoparticles.

Table 2. A column with references to studies with the systems described should be added. That will be useful for readers that need an in-depth study of each—the same for tables 3 and 4.

Author Response

Answers to added in the attachment.

Reviewer 2 Report

The manuscript submitted to evaluation for publication in Molecules is entitled: Versatile use of chitosan and hyaluronan in medicine. It s co-authored by K. Valachova & L. Soltes. It is a review that covers the essential applications of two major polysaccharides: chitosan and hyaluronic acid in its formulation as hyaluronan.

The authors address the application of these two polysaccharides in the following field of medical applications : (1) Viscosupplementation ; (2) Viscoprotection ; (3) Drug delivery ; (4) Dentistry ; (5) Cosmetics ; (6) Wound healing. The form of the manuscript is essentially descriptive, without any significant willingness to explain the relationship between the (macro)molecular structures and the functional properties underlying the medical applications. It might be considered awkward not to find any structural representations of the two polysaccharides, in a journal named « Molecule. »  

Points to consider :

Point 1. The address of the author is incomplete, as the name of the city is omitted. It could be appropriate to mention the institution: Slovak Academy of Sciences.

Point 2. Keywords. Without being wrong, the term « polycarbohydrates», is rarely unused.

Point 3. Line 47-48. Wrong assertion. Chitosan has indeed been promoted as an anti-obesity agent for a while. However, such an allegation has been dismissed for safety reasons. Therefore, the statement should be changed or accompanied by further explanation.

Point 4. Line 56-57. There is no reason the « write formerly named » HA is indeed a member of the glycosaminoglycan polysaccharides. In the paragraph, it could be appropriate to mention the different degrees of polymerization in correspondence to the molecular weight.

For macromolecules such as polysaccharide, the molecular weight, as expressed in Dalton or kDa. is preferred to the molecular mass.

Point 5. Line 92. Why the degradation of HA by hyaluronidases is not mentioned by the authors ?

Point 6. Line 93-94. The dual features of HA, having either a visco or an elastic behaviour as a function of the frequency it is submitted is a fascinating property. Some related documentation should be added to this section.

Point 7. Table 1. second line : should be enhancing

Point 8. Many of the functional properties of HA and Chitosan are related to such structural descriptors as: Degree of Polymerization (for both HA and Chitosan) and Degree of Acetylation for Chitosan). It would be interesting for the readers of « Molecule » to have such descriptors indicated concerning the medical application.

Point 9 ……When it comes to other applications, the authors should do their best to present which chemical, physico-chemical, …. modification of the native structure explain the functional property and derived application.

Having a pictorial representation of the molecular structures of chitosan and HA, along with an appropriate description of the key points explaining some of these properties would certainly enhance the value of the work.

Point 10. Why do the authors mention their affiliation to the Japan Society of Glycoforum ?

Author Response

Answers are added in the attachment.

Reviewer 3 Report

This narrative review is under the scope of this journal; the topic is relevant for readers, and this research deals with potentially significant knowledge to the field.

  • However, there are some concerns about the present manuscript: 

Title

  • I suggest removing “*”.

Abstract

  • In the abstract, the authors must describe the Authors should identify in the abstract the main points to be developed in the manuscript.
  • M&M - Identified how you made selected the articles?
  • M&M – Also Identified the time used for this selection?
  • M&M – what were the words used for this selection?

Manuscript

  • The description of the article search can be improved.Through the search, keywords and search engines, and time interval of this research.
  • Dentistry – Line 297, Palma et al. (DOI:1016/j.joen.2013.10.023) investigated in an animal study the usage of HA/chitosan scaffolds for dental pulp regeneration. But also use hyaluronic acid with pectin applied in endodontic regenerative procedures.
  • Tables- add a column with the authors and the references

References

  • The references are not standardized. Reference number 88.
  • The titles of references have a different format, the title of the article is written in capital letters at the beginning of words, others only in lower case.

Author Response

Answers are added in the attachment.

Round 2

Reviewer 3 Report

This research is under the scope of this journal; the topic is interesting for readers and this research deals with potentially significant knowledge to the field and an open new way for future studies.

The authors improved the quality of the manuscript after the reviewer's indications. Congratulations!!

Please add reference number 78 in the manuscript to the References,  (Palma, P.J.; Ramos, J.C.; Martins, J.B.; Diogenes, A.; Figueiredo, M.H.; Ferreira, P.; Viegas, C.; Santos, J.M. Histologic evaluation of regenerative endodontic procedures with the use of chitosan scaffolds in immature dog teeth with apical periodontitis. J. Endod. 2017, 43, 1279–1287. DOI:https://doi.org/10.1016/j.joen.2017.03.005).